# At What Age Could Screening for Familial Retinoblastoma Be Discontinued? A Systematic Review

**DOI:** 10.3390/cancers13081942

**Published:** 2021-04-17

**Authors:** Milo van Hoefen Wijsard, Saskia H. Serné, René H. Otten, Machteld I. Bosscha, Charlotte J. Dommering, Armida W. Fabius, Annette C. Moll

**Affiliations:** 1Department of Ophthalmology, Amsterdam UMC, Vrije Universiteit Amsterdam, Cancer Center Amsterdam, 1081 HV Amsterdam, The Netherlands; s.serne@amsterdamumc.nl (S.H.S.); m.bosscha@amsterdamumc.nl (M.I.B.); a.fabius@amsterdamumc.nl (A.W.F.); a.moll@amsterdamumc.nl (A.C.M.); 2Medical Library, Vrije Universiteit Amsterdam, 1081 HV Amsterdam, The Netherlands; r.otten@vu.nl; 3Department of Clinical Genetics, Amsterdam UMC, Vrije Universiteit Amsterdam, 1081 HV Amsterdam, The Netherlands; cj.dommering@amsterdamumc.nl

**Keywords:** familial retinoblastoma, age at diagnosis, screening

## Abstract

**Simple Summary:**

Offspring of patients with heritable retinoblastoma have a high risk of developing retinoblastoma themselves. Ophthalmological screening from birth in at-risk children ensures early detection and increases survival. As not every at-risk child develops retinoblastoma it should be determined until what age familial retinoblastoma can occur, so that ophthalmological screening could safely be stopped at that age. Extended screening beyond this age would result in unnecessary patient burden and costs. By systematically searching the medical literature for latest age at diagnosis in this population, we ascertained that among adequately screened patients the oldest age at diagnosis is 48 months. Therefore, screening for familial retinoblastoma can safely be stopped at four years of age.

**Abstract:**

The aim of this systematic review is to assess the latest age at diagnosis for detection of familial retinoblastoma in order to evaluate at what age screening of at-risk children could be discontinued. Extended screening beyond this age would result in unnecessary patient burden and costs. However, discontinuing screening prematurely would have the adverse effect of missing tumors. We performed a literature search (PubMed, Embase, CINAHL and the Cochrane Library) up until February of 2021 and systematically included studies where patients had a family history of retinoblastoma, a known age at diagnosis, and who were ophthalmologically screened for retinoblastoma from birth. A total of 176 familial retinoblastoma patients from 17 studies were included in this review. Based on 48 months of age being the latest age of diagnosis, ophthalmological screening for familial retinoblastoma could safely be discontinued at age four years.

## 1. Introduction

Retinoblastoma is the most common intraocular malignancy in children, which forms when both *RB1* alleles have been mutated in a susceptible developing retinal cell, leading to an inactivation of the retinoblastoma tumor suppressor [1]. Since retinoblastoma only develops in primitive retinal cells, which disappear a few years after birth, retinoblastoma rarely arises at an older age. Almost half of retinoblastoma patients have the heritable variant of disease, where subjects carry a germline *RB1* mutation, which can either arise de novo (sporadic) or is inherited from a parent [2]. Inherited disease is called familial retinoblastoma. Due to autosomal dominant inheritance and high penetrance of the inherited *RB1* mutation, offspring who carry the familial mutation have a highly increased risk of developing retinoblastoma. Genetic testing for *RB1* mutations in offspring aids in risk assessment but is not conclusive in all cases. Ophthalmological screening is crucial in children with a proven inherited *RB1* mutation and all offspring in families where the causative retinoblastoma mutation cannot be detected, hereafter named at-risk children. Early and frequent fundus screening by a dedicated ophthalmologist, preferably under general anesthesia, of at-risk children with a positive family history is the internationally accepted convention for retinoblastoma detection. Early diagnosis increases both survival and outcomes of sight [2] and could diminish the need for intensive treatment, such as chemotherapy, radiotherapy and enucleation [3]. Subsequent ophthalmological screening for retinoblastoma when disease is not apparent at birth presents a challenge, the risk of tumor development must be weighed against the consequences of general anesthesia, while also taking into careful consideration the psychological burden of repeated examinations in a young child and its parents, and the burden on healthcare systems including substantial costs.

Since genetic testing is not conclusive in all cases, is not widely available for all patients around the world, and since not all carriers of a germline mutation develop a tumor, it is important to determine the age at which tumors do not develop anymore. When no tumors have developed in an at-risk child, ophthalmological screening for familial retinoblastoma could be stopped at that age. Currently, there are conflicting recommendations in the literature until up to what age children at risk for familial retinoblastoma should be screened. In many countries/centers, at-risk children are screened until the age of four. This is in line with recommendations based in part on a previous study conducted in our institute over a 50 year period showing no children developing familial retinoblastoma after the age of four when properly screened from birth by a dedicated Ophthalmologist [4,5].

Recently published guidelines for screening retinoblastoma families, in contrast, recommend screening at-risk children up until the age of seven [6]. These guidelines were the result of consensus during an expert meeting in the United States, however they do not clearly state the scientific basis for this prolonged screening advice. To determine the age at which screening for familial retinoblastoma could be discontinued we systematically searched the literature for latest age of detection of a primary retinoblastoma tumor in patients who are at risk for familial retinoblastoma. To ensure that reported cases could not have been diagnosed much sooner, overestimating the latest age, we only included at-risk children who were screened from birth. 

## 2. Materials and Methods

### 2.1. Search Strategy

A literature search was performed by two authors (R.H.O. and S.H.S.), based on the Preferred Reporting Items for Systematic Reviews and Meta-Analysis (PRISMA) statement [7]. To identify all relevant publications on the age of diagnosis for familial retinoblastoma we systematically searched PubMed, EMBASE, CINAHL (via Ebsco) and The Cochrane Library (via Wiley) from inception to February 2021. Search terms included controlled terms (MeSH in PubMed and Emtree in Embase) as well as free text terms in The Cochrane Library. Publication language was restricted to English. Search terms expressing ‘hereditary retinoblastoma’ were used in combination with ‘screening’ and search terms comprising ‘age’. The full search strategies for all databases can be found in the Appendix A.

### 2.2. Selection Criteria and Data Extraction

Titles and abstracts were reviewed independently by two authors (S.H.S. and M.H.W.) and for relevant publications, full-text reports were individually evaluated by both. In case of disagreement, consensus was reached through discussion or by consulting a third author (A.C.M). Eligible studies included patients with a known family history of retinoblastoma whose age of detection of the first primary tumor is known. True age of diagnosis was researched by limiting inclusion to studies in which patients at risk were (adequately) screened for retinoblastoma from birth onwards, ensuring that age of detection was as soon as clinically possible. Screening performed within the first four weeks after birth was assessed as being screened from birth. Since we evaluate latest age at detection, studies where only a mean or median age at diagnosis was reported were excluded. All selected studies were evaluated for eligible patients. For each selected study general characteristics and data on familial retinoblastoma patients were extracted: number of patients with familial retinoblastoma, methods of screening, and (range of the) age of detection of the first primary tumor. Where necessary, age of detection of the first primary tumor was converted into months. When not reported in the publication itself, if possible, the mean and median age at diagnosis were calculated for each study.

A combined median age at diagnosis was determined for familial patients from included studies where individual diagnostic age could be extracted from the data. A combined mean age at diagnosis was estimated from the same data combined with studies without individual diagnostic age but where a mean age was reported. Prenatal diagnoses were counted as diagnosis at birth.

## 3. Results

### 3.1. Included Studies and Patients

A total of 1001 studies matching our search criteria were identified. A flowchart of the selection process is depicted in Figure 1. After title and abstract screening, 210 full-text articles were assessed for eligibility. Many studies that include familial retinoblastoma cases only report mean and/or median age or make no distinction between familial and non-familial cases when reporting latest diagnosis or the range of age at diagnosis. Therefore, these studies could not be included. In addition, several studies were identified in which the range in age of familial cases was reported, but it was unclear whether patients were adequately screened from birth. Ultimately, 17 studies with eligible patients were identified and included in this systematic review [4,5,8,9,10,11,12,13,14,15,16,17,18,19,20,21,22].

All 17 studies were conducted between 1976 and 2020, encompassing patient data from 1914 onwards. Most (nine) were retrospective cohort studies [4,5,9,10,11,12,13,14,15], the eight remaining studies consisted of case reports [16,17,18,19,20,21,22]. Table 1 shows the extracted data for the included studies.

From 17 studies, a total of 176 eligible patients with familial retinoblastoma were included. Not all familial retinoblastoma cases from every study could be included, mostly because screening intervals differed per patient and not all cases were diagnosed through screening. A previous study identifying the latest age at diagnosis in the Dutch retinoblastoma population to determine until what age screening is needed reports that 50 of 75 familial patients were screened from birth [4]. In a retrospective overview of familial retinoblastoma patients with a normal eye exam on first examination only thirteen of twenty-one patients were screened within the first month, excluding eight from our analyses [13]. In a comparison study which demonstrated the importance of early and regular fundus examinations under general anesthesia, only the 16 patients with systematic fundus screening according to their proposed guidelines were included [5]. Retrospective analyses of clinical outcomes in a large cohort of unilateral retinoblastoma patients reported three familial cases of which two were screened from birth [15], where in a case series on trilateral retinoblastoma only one of three cases was evidently screened for familial retinoblastoma [21]. Lastly, in a small study assessing prenatal retinoblastoma lesion detection using screening MRI, one of five cases remained disease free at end of follow up and was not included in our study [14].

Three included studies from Abramson and colleagues report on retinoblastoma patients from the same US ophthalmic oncology center. Albeit either focusing solely on unilateral retinoblastoma patients (*n* = 22 included) [9], bilateral retinoblastoma patients (*n* = 22 included) [10], or with previous normal eye examination (*n* = 13 included) [13], overlap in reported patients between these studies cannot be excluded and is probable in at least eight cases between the latter two studies. Because individual patients could not be identified and possible overlap does not impact our primary outcome measure, latest age at diagnosis, all cases were still included in Table 1. The same goes for the two included studies on familial retinoblastoma conducted within the Dutch nationwide Rb registry; since study periods 1945−1998 [4] and 1992−2004 [12] overlap, it is likely some cases are included twice.

### 3.2. Age at Diagnosis

Most studies reported the age at retinoblastoma diagnosis as an outcome in text or table where in some studies the range had to be extracted from a figure [5,10,11]. Where necessary, the (range of) age of detection of the first primary tumor was converted into months. Sometimes mean and median age at diagnosis were presented in the study, otherwise they were calculated based on reported individual patient data (see Table 1). In one of the included studies, the age for one of two eligible patients did not match between main text and Appendix A (diagnostic age of three months in text versus seven months in Appendix A) [15]. Since the purpose of this review is to identify the oldest patient in whom familial retinoblastoma has been diagnosed by screening from birth, and it cannot by established which of these two ages is correct, we have included the highest mentioned age of seven months in our review.

Within a total of 176 eligible familial patients, the age at diagnosis of first retinoblastoma tumor ranged from 0 to 48 months. The latest age at retinoblastoma diagnosis, 48 months, was reported in a retrospective review of familial Retinoblastoma in the Dutch nationwide cohort between 1945–1998. The study reports the second oldest diagnosis was at 24 months of age in two patients. In all 15 other included studies, the oldest patient was three years of age or younger at time of diagnosis, with most studies even reporting the oldest before the age of one year.

Within the 14 studies (*n* = 88) where a patient’s age was individually reported [8,10,11,12,13,14,15,16,17,18,19,20,21,22] the combined median age at diagnosis was 2.0 months of age (95% CI: 1.3–3.0 months). When also including one other study where the mean age was reported in the text [4], based on these 15 studies (*n* = 138), the combined mean age at diagnosis was 3.6 months of age (95% CI: 3.1–4.2 months).

### 3.3. Screening for Familial Retinoblastoma

Reporting on screening methods differed per study. Included retrospective cohort studies reported that at-risk children were screened at least within the first four weeks after birth, and examined regularly thereafter until the end of follow-up. Most cohorts did not specify their exact screening practices in these publications [4,10,11,12,13,14], while Rothschild and colleagues specifically compared different intensities of screening, recommending the widely recognized fundus screening practice of early screening within the first week after birth, every month up to 18 months of age, then every 3 months up to 4 years of age [5]. One of the included studies from Abramson and colleagues reports that most of the patients fundus examinations were performed at birth or shortly thereafter, with follow-up every six weeks [9]. Most included case series report intentional fundus screening from birth in at-risk cases, but describe no further screening practice due to early retinoblastoma diagnosis within the first months after birth [8,16,18,20,21,22], while in one case fundus evaluation and diagnosis of retinoblastoma was within the first month while not mentioning screening [17] and in another case report a child was diagnosed at 28 months after periodic monthly examination under general anesthesia until 2 years of age [19].

## 4. Discussion

We found that the latest age of retinoblastoma diagnosis for a child at risk for familial retinoblastoma who is screened from birth is four years of age. Moreover, all the other 175 familial patients (99%) identified in the literature were diagnosed within the first three years of life. Since all these patients underwent ophthalmological examination for retinoblastoma from within the first month after birth, these reported diagnostic ages are near to the true age of appearance. 

The oldest published age at initial retinoblastoma diagnosis at four years of age, was reported 20 years ago in a study from our own research group [4]. This study, appropriately titled ‘*At what age could screening for familial retinoblastoma be stopped?*’, bases its recommendation to screen until four years on a retrospective review of all familial retinoblastoma diagnoses in the Netherlands in a 53-year long period, 1945−1998. Fifty of seventy-five familial patients were assessed to have been ophthalmologically screened from birth with aforementioned latest diagnosis at four years. All other 49 patients were diagnosed at least two years earlier with second-latest diagnoses at 24 months of age for two familial patients. This gap in diagnostic age, coupled with all other familial patients from the 17 included studies being diagnosed at 28 months at the latest, makes this one patient diagnosed at 48 months an outlier. As this patient was diagnosed in an era were not yet every at-risk child in the Netherlands was screened by a dedicated Ophthalmologist specialized in retinoblastoma, or even under general anesthesia, it is fairly possible this diagnosis could have been made sooner with more adequate screening. More so, a follow-up study within the same Dutch cohort, included in this review, where in a later time period of 1992−2004 all 17 consequently screened familial retinoblastoma were evaluated showed a latest age at diagnosis of four months [12]. Thus, when disregarding the one outlier of 48 months, one could argue ophthalmological screening could even be safely stopped at three years of age.

Rothschild and colleagues’ screening guideline for retinoblastoma is based on retrospective comparison of screening intensity within their familial population (*n* = 59) from 1995 to 2004 [5]. Besides showing decreased necessity for enucleation and radiotherapy with significantly earlier age at diagnosis in the more thoroughly screened cases, their recommendation is also to systematically screen until age four. This, with seven months being the latest age at diagnosis of their 16 intensively screened patients included in this current review. Whereas 35 months of age is the latest for the 23 patients whose screening did not match the recommended interval, which is still a full year below the recommended four years. On the other hand, the oldest of their 20 non-screened familial patients, diagnosed due to clinical signs, was 57 months of age, illustrating these very late diagnoses occur only in non-screened patients. This is supported by the latest age at diagnosis in non-screened familial patients of 63 months of age in the abovementioned Dutch cohort [4].

Underscoring the importance of regular and systematic screening, Abramson and colleagues determined at what age familial patients with an initial normal eye exam developed retinoblastoma tumors [13]. In this study, 13 out of 21 patients with a normal eye examination within the first month, met the criteria of our review and were included, the oldest diagnosed at 14 months of age. Moreover, all eight other patients who had their first, initially normal eye exam after the first month of life were still diagnosed with retinoblastoma within 28 months of age. The same can be seen in other studies describing screened familial patients who had their first eye examination after the first month of life and still show a retinoblastoma diagnosis below the age of three years [23,24,25,26].

To our knowledge, this is the first systematic literature review to evaluate age at diagnosis in this population. By looking only at familial patients who are adequately screened for retinoblastoma from birth, we exclude patients who actually could have been diagnosed much earlier in life, preventing overestimating the true oldest age at diagnosis. As most studies do not specifically report the range of diagnostic age for their subset of familial patients, the amount of studies that could eventually be included in this systematic review is not overwhelming. Nevertheless, all 17 studies with 176 patients show that no properly screened patient was diagnosed after four years of age. Furthermore, although screening methods varied per study and outcomes may have differed if all patients were screened using the same techniques and schedule, harmonized screening would have likely produced younger overall age at diagnosis, and not an age at diagnosis beyond the oldest reported age. Additionally, three studies from the same US center and two from the same Dutch center were included in this review. While an apparent small overlap in patients may overstate the total number of included patients, it does not affect the latest age of diagnosis.

## 5. Conclusions

Based on this systematic review of the literature, the latest age of diagnosis in familial retinoblastoma patients who were screened from birth is four years. All other adequately screened patients we identified were diagnosed before three years of age. According to our findings, the risk of developing retinoblastoma past the age of four years in these at-risk children therefore seems minimal to none. In our opinion, when weighing the multidimensional burden of continuous medical examinations against this minimal risk, ophthalmological screening can safely be stopped at four years of age.

## Figures and Tables

**Figure 1 cancers-13-01942-f001:**
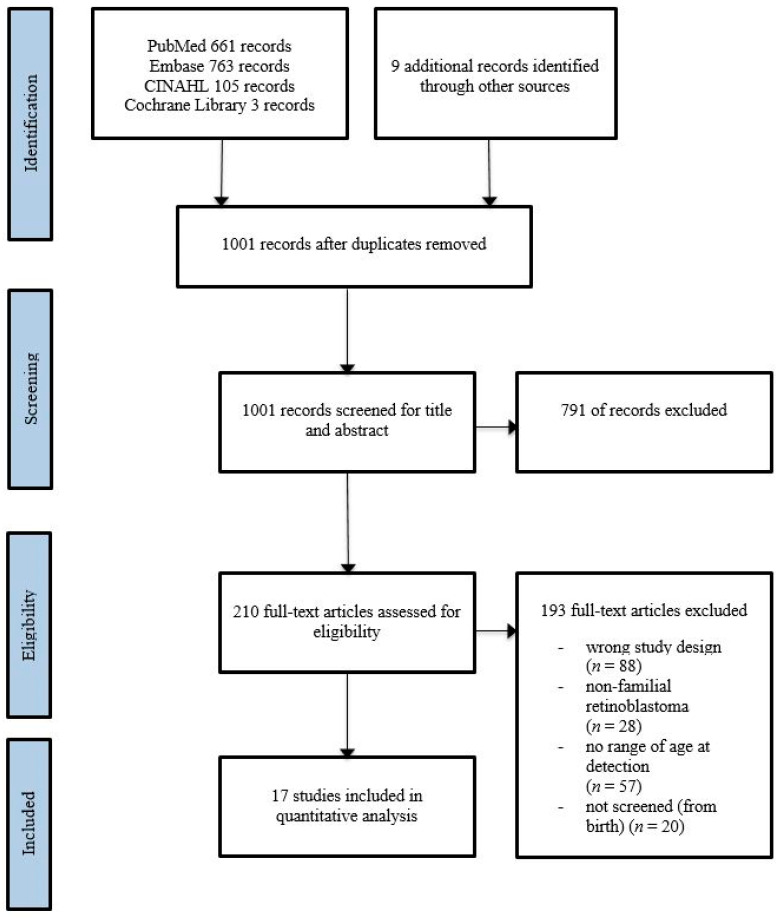
Flow diagram of the study inclusion process.

**Table 1 cancers-13-01942-t001:** Age at familial retinoblastoma diagnosis in patients screened from birth.

Author (Year)	Study Design	Years	At-Risk Children Screened for Familial Rb (*N*)	Ophthalmological Screening	Age at Rb Diagnosis (Months)		
				from Birth	Range	Mean	Median
Moll et al. (2000)	Retrospective cohort study	1945−1998	50	Yes	0−48.0	4.9	1.9
Abramson et al. (1982)	Retrospective cohort study	1914−1981	22	Mostly	0.5−28.0	N/A	2.5
Abramson et al. (1981)	Retrospective cohort study	1955−1980	22	Yes	0−11.0	3.8 *	3.0
Soliman et al. (2016)	Retrospective cohort study	1996−2014	20	Yes	0−6.5	2.4 *	2.0 *
Imhof et al. (2006)	Retrospective cohort study	1992−2004	17	Yes	0.5−4.0 ^†^	1.2 * ^†^	0.5 * ^†^
Rothschild et al. (2011)	Retrospective cohort study	1995−2004	16	Yes	0−7.0	N/A	0.0
Abramson et al. (1998)	Retrospective cohort study	1960−1990	13	Yes	0−14.0	4.8 *	3.5 *
Staffieri et al. (2015)	Retrospective cohort study	2008−2013	4	Yes	0−1.5	0.9 *	1.1 *
Mallipatna et al. (2009)	Retrospective cohort study	1988−2008	2	Yes	1.0−7.0	4.0	4.0
Fang et al. (2020)	Case report	2006−2019	1	Yes	0.3	-	-
Kaliki & Jajapuram (2019)	Case report	2016−2017	2	Yes	0.5−0.7	0.6	0.6
Yarovaya et al. (2017)	Case report	N/A	1	Yes	0.7	-	-
Shah et al. (2016)	Case report	N/A	1	Yes	0.3	-	-
Neriyanuri et al. (2015)	Case report	N/A	1	Yes	28.0	-	-
Pierro et al. (1993)	Case report	1989−1991	2	Yes	0.1−0.1 *	0.1 *	0.1 *
Holladay et al. (1991)	Case report	N/A	1	Yes	0	-	-
Howe et al. (1976)	Case report	1973	1	Yes	1.5	-	-

N/A = not applicable. Rb = retinoblastoma. Notes: ophthalmological screening for retinoblastoma within 4 weeks from birth was counted as screening from birth. Age at diagnosis is converted into months. Prenatal and diagnosis at birth is counted as 0 months. * Mean and/or median calculated based on data extracted from figure or table. ^†^ Age 1−2 weeks was counted as 2 weeks of age.

## Data Availability

Data is contained within the article and Appendix A.

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
