# Peer review of "At What Age Could Screening for Familial Retinoblastoma Be Discontinued? A Systematic Review"

_cancers, 2021, doi:10.3390/cancers13081942_

Round 1
Reviewer 1 Report
In this commentary, the authors address the question of until what age ophthalmological screening is required in individuals with familial retinoblastoma. They point out that a US expert meeting concluded to "recommend screening ... up until the age of seven" but that no objective evidence for this recommendation was provided.
Previously published observations on screening outcomes in individuals with familial retinoblastoma might serve as one source of such objective evidence. Consequently, the authors performed a thorough literature search. Their commentary is based on the results of this literature search. Specifically, in the whole dataset, 48 months was the latest age at diagnosis. From this finding, the authors infer that "ophthalmological screening for familial retinoblastoma could safely be discontinued at age four years" (lines 34-35).
This commentary is important because it points out that the US expert's recommendation appears to be too conservative by a wide margin given reported observations. However, the authors do not present the statistical inference results to justify their recommendation "ophthalmological screening for familial retinoblastoma could safely be discontinued at age four years." It is plausible that the risk of retinoblastoma in children with a heritable predisposition is low by age at four. However, scientific methods of inference are required to determine the level of confidence (error margin). Admittedly, a thorough statistical analysis, which may also include other independent variables, is beyond a commentary's scope.
In summary, this is an important commentary that will bring relief from the burden of examinations for children with very low or even absent risk.
remarks:
- The statement "Therefore, screening for familial retinoblastoma can safely be stopped/could safely be discontinued at 4 years of age." is not backed by proper statistical inference. Stating the factual along the line "among xx observations, no child with new retinoblastoma older than 48 months" is sufficient at this stage.
- line 140: cases
- Presentation of literature data: I wonder if it was possible to provide some graphs, e.g., stacked dot plots (https://eurekastatistics.com/r-flavours-of-stacked-dotplots/) to give an impression of the distribution of age at diagnosis.
- Overall, the manuscript might be shortened without losing information by getting rid of redundancies and repetitions, e.g., lines 221-229.
- Line 280: the argumentation is an example of circular reasoning.
Author Response
Response:
We thank the reviewer for these comments and suggestions. And thank the reviewer for acknowledging the importance of reducing burdenous screening as much as possible in this paediatric population.
We wholeheartedly agree that confidence intervals aid risk assessment, and that additional analyses including independent variables indeed reach beyond the scope of this Commentary. Also, the data available in the literature does not allow this for the measure we are interested in (i.e. range; latest age at diagnosis) because for half of the identified cases no individual age at diagnosis, but only the mean age and/or range was reported. We identified only one patient diagnosed at the age of 4 years (in our own cohort) while all other patients in literature were diagnosed before the age of three years. As this one patient was diagnosed in an era were not yet every at-risk child in the Netherlands was screened by a dedicated Ophthalmologist specialized in retinoblastoma, or even under general anaesthesia, it is likely this diagnosis could have been made sooner with more adequate screening. With both this and the burden of screening in mind we would advise to stop ophthalmological screening at age 4.
We revised our conclusion to better reflect that this is our stance while still mentioning the slight uncertainty of our conclusion, adjusted in bold.
“Based on this systematic review of the literature, the latest age of diagnosis in familial retinoblastoma patients who were screened from birth is 4 years. All other adequately screened patients we identified were diagnosed before 3 years of age. According to our findings, the risk of developing retinoblastoma past the age of 4 years in these at-risk children therefore seems minimal to none. In our opinion, when weighing the multidimensional burden of continuous medical examinations against this minimal risk, ophthalmological screening can safely be stopped at 4 years of age.” (line 304 - 310).
Further, confidence intervals are now reported for the combined mean and median age at diagnosis:
- “combined median age at diagnosis was 2.0 months of age (95% CI: 1.3 - 3.0 months)” (line 196)
- “combined mean age at diagnosis was 3.6 months of age (95% CI: 3.1 – 4.2 months)” (line 199)
2) line 140: cases
Response: thank you for pointing this out. Changed to ‘cases’ (line 142)
3) Presentation of literature data: I wonder if it was possible to provide some graphs, e.g., stacked dot plots (https://eurekastatistics.com/r-flavours-of-stacked-dotplots/) to give an impression of the distribution of age at diagnosis.
Response: thank you for this suggestion. We refrained from reporting the distribution of age at diagnosis other than the combined mean and median for the cases where individual diagnostic age was reported. Since only half of all included cases have individual age at diagnosis reported in the respective study (and not just mean age and/or range), showing this distribution would not properly reflect the true distribution and might give the wrong impression of all 176 included familial cases. If the editor would like us to present a figure showing the distribution in the cases for which this is possible, we would be happy to do so.
4) Overall, the manuscript might be shortened without losing information by getting rid of redundancies and repetitions, e.g., lines 221-229.
Response: Thank you for pointing this out. We agree that the motivation for conducting this study was already properly addressed in the introduction. The paragraph of the discussion Reviewer 1 refers to has been removed altogether.
5) Line 280: the argumentation is an example of circular reasoning.
Response: Thank you for the opportunity to clarify this. Although we do not think this qualifies as a circular reasoning we changed the line to prevent any confusion. Changes in bold: Nevertheless, all 17 studies with 176 patients show that no properly screened patient was diagnosed after 4 years of age. (line 294)

Reviewer 2 Report
The manuscript entitled ‘At what age could screening for familial retinoblastoma be dis-continued? A systematic review’ by van Hoefen Wijsard M et al is a systematic review to assess the latest age at diagnosis for detection of familial retinoblastoma so to evaluate a recommended age of at-risk children at which a screening could be discontinued. The described results have been obtained following a literature search on PubMed, Embase, CINAHL and the Cochrane Library and the conclusions report that the screening can safely be stopped at 4 years age.
First of all, I would like to underline that, among the others, a recent publication indicates that the percentage of familial history of retinoblastoma, as compared to the percentage of non-familial history of the disease, is low thus allowing the detection of only a minority of cases on a global scale (Jama Oncology, 2020;6(5):685-695). Moreover, even if there is quite a large consensus on the median age of diagnosis around 48 months old, some publications indicate that retinoblastoma is detectable also in children and adolescents > 5 years and recommend the screening at least until age 5 (Ries LAG, et al (eds).Cancer Incidence and Survival among Children and Adolescents: United StatesSEER Program 1975-1995, National Cancer Institute, SEER Program. NIHPub. No. 99-4649. Bethesda, MD, 1999; Clinical Cancer Res, 2017;23(13):e98-e106; Int. J. Cancer 2021;1-7). In addition, the literature clearly indicates that the analysis of retinoblastoma at the time of diagnosis revealed important differences in presentation among patients from different countries, depending on their national income level (Jama Oncology, 2020;6(5):685-695; Br J Ophtalmol 2020;0:1-9; Lancet Oncol 2018;19:e252-266).
Considering this last aspect, it would be particularly important if the authors further analyze the data by countries with different national income level. Also, it would be interesting to know, whenever available, whether there is a relation among the age at diagnosis and 1) the laterality at diagnosis, 2) the stage of tumor detected, 3) the presence of distant metastases. Moreover, in the light of the published literature, the data should be more deeply discussed.
Author Response
Response: We thank the reviewer for the comments and suggestions, and for raising some important issues about disparity in retinoblastoma care around the world.
We think it is important to emphasize that our study focusses solely on inherited familial retinoblastoma. More specifically, we look at screening of children at risk for inherited familial retinoblastoma because of a known positive family history (so with one parent already identified with retinoblastoma). While we are aware of the mean age of diagnosis of retinoblastoma in its entirety, and agree that patients with sporadic non-heritable disease have presented and been diagnosed with retinoblastoma well after 5 years, we specifically look only at inherited familial retinoblastoma in our systematic review and make no statements on age of diagnosis of non-familial retinoblastoma. We thank the reviewer for emphasizing the observed differences in diagnostic age of retinoblastoma between countries with different national income levels and we underscore the importance of this issue. As the studies that the reviewer cites also conclude, the difference in age of diagnosis between these countries is mainly caused by late presentation and later appropriate diagnosis in countries with less financial resources, and again this is when looking at retinoblastoma diagnoses in general and not specifically familial retinoblastoma. Since we focus on screening children with positive family history from birth, late presentation is not an issue as these children are already known by the physician. We do not expect a difference in age at diagnosis of familial retinoblastoma between different countries when at-risk children are screened from birth according to the same protocol. We do think that the conclusion we draw from our systematic search of the literature, namely that all adequately screened familial retinoblastoma patients identified were diagnosed before 4 years of age and this is therefore a safe cut-off point for screening, can be applied in around the world as long as at-risk children are properly screened until the cut-off point of 4 years of age.

Round 2
Reviewer 2 Report
I thank the authors for their reply. I think that their systematic review would have been more interesting by adding further analyses of the literature they extrapolated, as suggested. However, I take into account that the focus of their systematic review is only limited on the inherited familial retinoblastoma and on the cut-off children age at which it is safe to stop the screening.